# The impact of climate change on the agriculture and the economy of Southern Gaul: New perspectives of agent-based modelling

**Nicolas Bernigaud**[1]*, **Alberte Bondeau**[2], **Joël Guiot**[1], **Frédérique Bertoncello**[3], **Marie-Jeanne Ouriachi**[3], **Laurent Bouby**[4], **Philippe Leveau**[5], **Loup Bernard**[6], **Delphine Isoardi**[5]

**1** CNRS, IRD, INRA, Coll France, CEREGE, Aix-Marseille Université, Aix-en-Provence, France, **2** Institut Méditerranéen de Biodiversité et d'Écologie Marine et Continentale (IMBE), Aix-Marseille Université, CNRS, IRD, Avignon Université, Aix-en-Provence, France, **3** CNRS, CEPAM, Côte-d'Azur Université, Nice, France, **4** CNRS, Institut des Sciences de l'Evolution-Montpellier (ISEM), Montpellier Université, Montpellier, France, **5** CNRS, Centre Camille Julian (CCJ), Aix-Marseille Université, Aix-en-Provence, France, **6** CNRS, Archimède UMR, Strasbourg Université, Haute-Alsace Université, Strasbourg, France

* bernigaud.nico@orange.fr

**Data Availability Statement:** The Netlogo model ROMCLIM with the other files are available in

## Abstract

What impact did the Roman Climate Optimum (RCO) and the Late Antique Little Ice Age (LALIA) have on the rise and fall of the Roman Empire? Our article presents an agent-based modelling (ABM) approach developed to evaluate the impact of climate change on the profitability of vineyards, olive groves, and grain farms in Southern Gaul, which were the main source of wealth in the roman period. This ABM simulates an agroecosystem model which processes potential agricultural yield values from paleoclimatic data. The model calculates the revenues made by agricultural exploitations from the sale of crops whose annual volumes vary according to climate and market prices. The potential profits made by the different agricultural exploitations are calculated by deducting from the income the operating and transportation costs. We conclude that the warm and wet climate of the Roman period may have had an extremely beneficial effect on the profitability of wine and olive farms between the 2nd century BCE and the 3rd century CE, but a more modest effect on grain production. Subsequently, there is a significant decrease in the potential profitability of farms during the Late Antique Little Ice Age (4th-7th century CE). Comparing the results of our model with archaeological data enables us to discuss the impact of these climatic fluctuations on the agricultural and economic growth, and then their subsequent recession in Southern Gaul from the beginning to the end of antiquity.

## Introduction

The evidence of a warming climate between the 3rd century BCE and the 2nd century CE based on various continental and oceanic paleoclimate proxies [1–3] has led to questions about its

GitHub https://github.com/Bernigaud2021/
ROMCLIM

**Funding:** JG and his team has received funding from Excellence Initiative of Aix-498 Marseille University - A*MIDEX, a French "Investissements d'Avenir" programme, through the 499 RDMed project and Labex OT-Med project (project ANR-11- LABEX-0061). This work was also supported by the ANR Project MICA (dir. L. Bouby & N. Bernigaud), funded by the French National Research Agency (Agence Nationale de la Recherche) [grant number ANR-22-CE27-0026]. The funders had no role in study design, data collection and analysis, decision to publish, or preparation of the manuscript.

**Competing interests:** The authors have declared that no competing interests exist.

impact on ancient societies. While the beneficial role of this Roman Climate Optimum (RCO) on agriculture in the Roman Empire has recently been evoked, as well as the negative effect of the Late Antique Little Ice Age [4], it is still difficult to measure with certainty the effects of these climatic fluctuations on agricultural yields and the economy based on textual and archaeological sources alone, which are often very scanty and incomplete or difficult to interpret [5].

The application of agent-based model to archaeology, which provides a new exploratory tool, now makes it possible to overcome these difficulties. It is indeed possible to test multiple hypotheses on the interactions between societies and their environment, independently of the quantity or quality of the data available. In archaeology, this type of modelling has been used for some years to deal with questions related to the demography of ancient societies, agricultural strategies, carrying capacity, and settlement dynamics (details in I, 4).

In this paper, we present our agent-based model, ROMCLIM, which we designed to test the impact of climatic changes between the Iron Age and late antiquity (6th century BCE - 7th century CE) on the potential profitability of farms in Southern Gaul. But this model is also a tool with predictive ability for reconstructing the geography of crops, still partially known by archaeological data. One of the interests of ROMCLIM is to emulate and simplify a complex agro-ecosystemic model (LPJmL) to simulate impact of climate change on crops, which is an important point to understand relationship between climate and ancient societies. The geographical area considered corresponds to most of the ancient Roman province of Gallia Narbonensis, which covered Provence and Languedoc, as well as the middle Rhone valley.

The model can simulate the profits of different types of farms by calculating the average potential agricultural yields for vines, olives, and cereals (wheat/barley), which were the most cultivated crops as of the Roman period. This "Mediterranean triad" was indeed the foundation of agriculture and food in most of the Roman Empire, and therefore its primary source of wealth.

## The study area

The study area corresponds to most of the province of Gallia Narbonensis, conquered by Rome in the 120s BC. Located in Southern France, this area (41˚52' N-44˚55' N latitude, 2˚21' E-7˚12' E longitude) encompasses all or part of the current administrative regions of Occitanie, Provence-Alpes-Côte-d'Azur (PACA) and Auvergne-Rhône-Alpes. It is limited to the longitude of Antibes to the east, that of Toulouse to the west and the latitude of Valence to the north. From a geomorphological point of view, this area surrounded by the Alps, the Massif Central and the eastern part of the Pyrenees, is formed by coastal plains and hilly and mountainous hinterland. The Rhone Valley and its delta separate to the west and east the historical regions of Languedoc and Provence.

The Mediterranean climate of this area is characterized by significant annual sunshine (2500–2900 hours/year), high summer temperatures and a low number of frost days (0–40), compared to the rest of France. Precipitation is low in the coastal plains (500–600 mm/year), but much more abundant in the mountain ranges, especially the Cevennes (1200–1700 mm/ year) which receive very intense rainfall in autumn (Fig 1).

## State of the art

### Agricultural production according to historical and archaeological sources

**Vine and olive tree.** According to the Gallo-Roman historian Pompey Trogue, the Greeks taught the Gallic to cultivate vines and olive trees (Justin, *Epitoma historiarum Philippicarum Pompei Trogi*, XLIII, 4, 2). Viticulture was probably introduced to Gaul by the Greeks after the foundation of Marseilles around 600 BCE, as evidenced by the discovery of vine planting pits

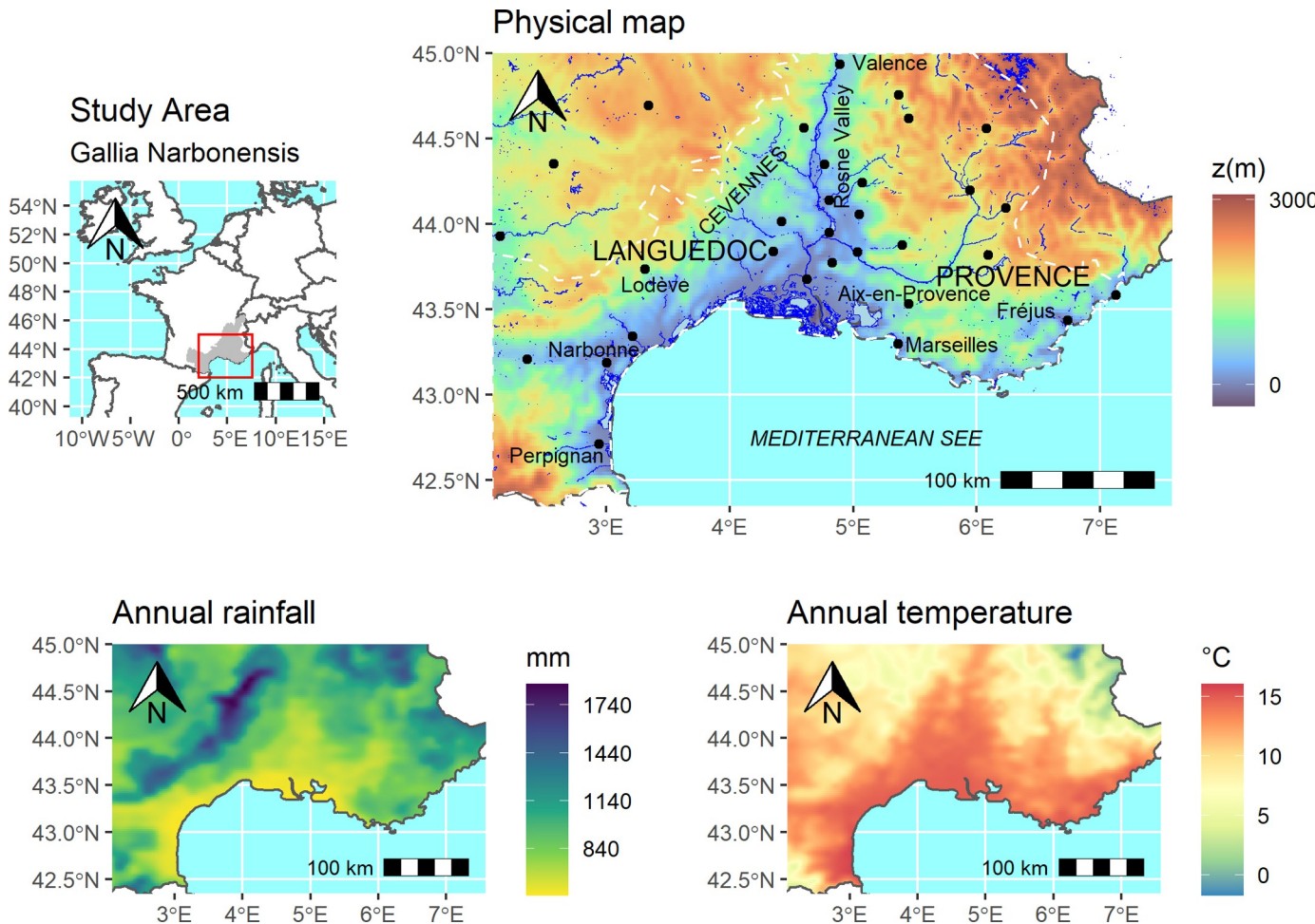

**Fig 1. Location of the study area, physical and climate maps.** Annual rainfall and temperature maps were processed with data extracted from ALADIN Model.

on the territory of the Phocean colony [6]. During the Iron Age, Marseille wine was sold in Southern Gaul, as indicated by the discoveries of Massaliot amphorae that contained this wine [7]. Other Etruscan and Greek *emporia* along the Mediterranean coast have certainly also developed viticulture locally. In the protohistoric agglomeration of *Lattara*-Lattes (Hérault) carpological studies and biochemical analyses clearly attest to its intensification from the 3rd century BCE [8, 9]. Despite an importance that appears less and less negligible, this protohistoric viticulture seems to have remained confined during the Iron Age to the Mediterranean coastal strip. The volume of production was obviously insufficient to satisfy the consumption of the rest of Gaul where Italic wine was consumed (as indicated by the discoveries of amphorae of the Dressel 1 type).

After the Roman conquest, viticulture in Gaul experienced a gradual intensification from the 1st century BCE. At the beginning of our era the Greek geographer Strabo (*Geography*, IV, I, 2) described the landscape of this province as like that of Italy because of its crops: vineyards, olive trees and fig trees flourished as far as the Cevennes, before becoming scarce beyond this limit. Archaeological investigations have largely confirmed the importance of viticulture in Languedoc and Provence, as evidenced by the discovery of the remains of wine-growing establishments, wine amphorae workshops, winegrowers' tools, and grape seeds [10–14] (Fig 2).

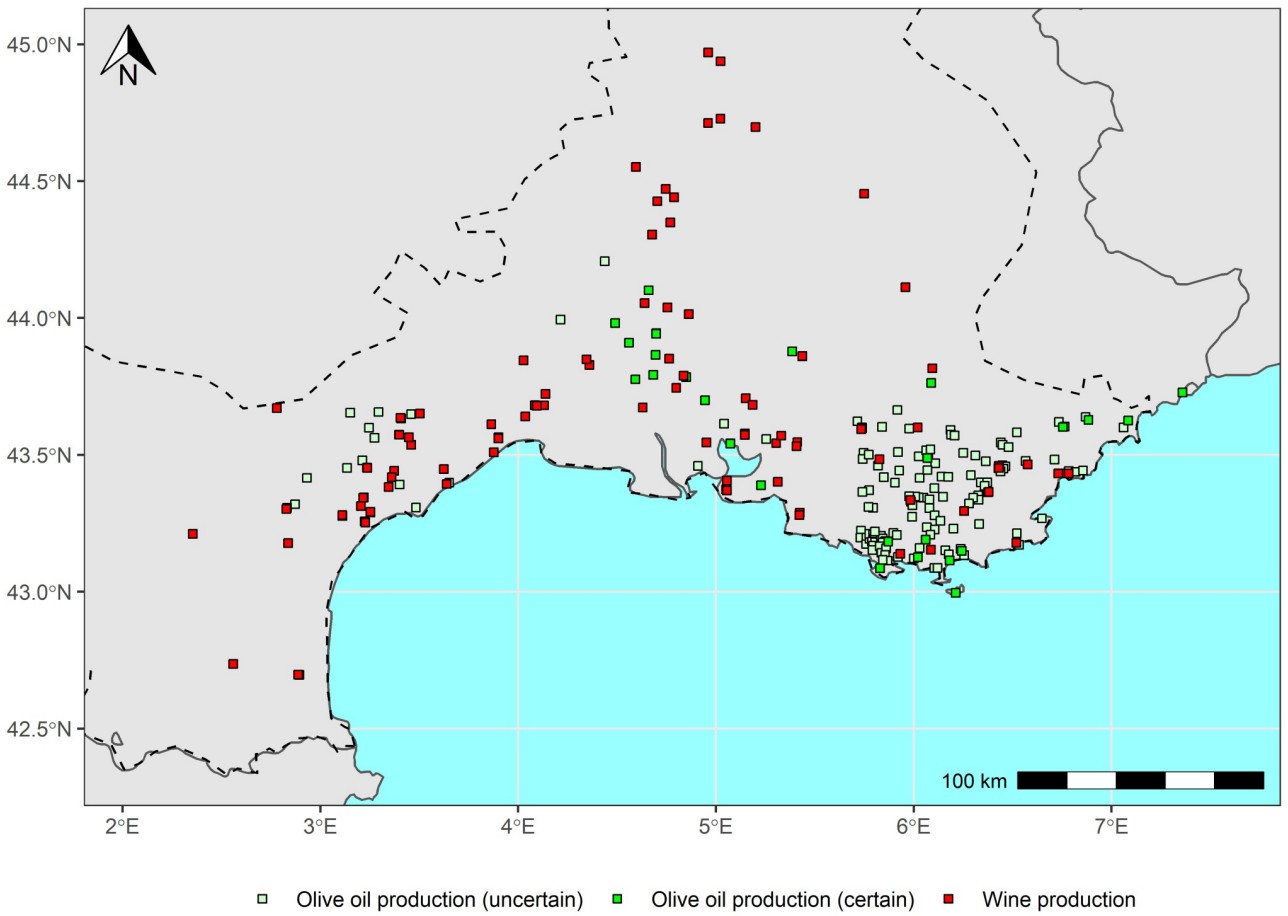

**Fig 2. Map of archaeological data referring to wine and olive oil production in late iron age and roman period (data mainly extracted from Garcia 1992 and Brun 2005).**

After reaching its climax in the 2nd century CE, it began to decline in Narbonne from the 3rd century CE but nevertheless continued to spread throughout Gaul [13], as we recently published a paper on this topic with numerous archaeological data [15].

Regarding olive growing, palynological and carpological studies also place towards the end of the 3rd century CE its development in Provence [16, 17]. At the beginning of our era, Strabo evoked his presence in the region of Marseilles (*Geography*, IV, I, 5) and more broadly in Gallia Narbonensis, as already mentioned. In the 1980s, the first regional archaeological syntheses identified in Provence and Languedoc presses and millstones linked to an olive activity of some importance [13, 18, 19]. However, this interpretation has since been questioned. It was objected that these installations and tools could also have been used to press grapes for viticulture. On the other hand, the strong representation of Betic amphorae on the Gallo-Roman sites of Southern Gaul indicates that olive oil was consumed there, produced in the south of the Iberian Peninsula. These observations therefore led to the conclusion that the importance of ancient olive growing in Narbonne should now be seriously revised downwards [13, 20]. It is now even considered that it should have been of little importance, but the question is not definitively settled. Recently, carpological and anthracological analyses have revealed in Roussillon the existence of olive growing during the High Empire [21]. The progress of

archaeobotanical studies will undoubtedly lead in the future to reassess (based on less questionable data than those of press installations) its importance in Southern Gaul.

**Grain cultivation.** During the Iron Age, the cultivation of cereals and pulses ensured food production throughout Gaul. Carpological studies have determined that clothed barley was the most important crop. Different varieties of millets and wheats were also grown, such as einkorn, emmer but especially naked wheat, the cultivation of which took on increasing importance during the Second Iron Age [22, 23]. This phenomenon is perhaps related to the probable development of a commercial cereal crop intended to supply the Greek city of Marseilles, whose arid territory was rather suitable for the cultivation of vines and olive trees. Southern Gallic and Phoceans would then have exchanged cereals for wine [24, 25].

In the Gallo-Roman period, the importance of cereal cultivation in Narbonne is still difficult to determine compared to other crops. First, written sources provide little information on this subject. An inscription painted on a small amphora formerly discovered in the port of Marseilles provides, for example, the very punctual testimony of a cargo of barley imported into the Phocean city from the country of Cavares (current department of Vaucluse) [26]. Carpological studies–still few on sites from the Roman period–highlight among the various cereals cultivated, the preponderance of barley and naked wheat, already cultivated before the Roman conquest [23, 27]. Finally, although archaeological investigations conclude that viticulture is omnipresent in Gallia Narbonensis [28], discoveries of Gallo-Roman structures linked to cereal production (granaries, silos, mills, etc.) are still rare [29]. This observation raises the question of whether there is an archaeological bias or whether cereal cultivation had really lost importance in the Roman period, to the benefit of viticulture.

## What rural economy in roman times in Southern Gaul? Autarkic or trade-oriented agriculture?

The ROMCLIM model presented in this article makes it possible to explore the hypothesis of commercial agriculture spread over the entire study area, but this vision of the rural economy is not shared by all historians and archaeologists. Did the estates of Southern Gaul practice an autarkic agriculture or were they turned towards commercial production? Although the vision of an ancient autarkic economy outlined by the American historian Finley [30] has long been challenged by eminent historians of Roman economics [31–35], Gallo-Roman villae of Southern Gaul (and elsewhere) are still often presented today as areas where polyculture was practiced primarily for the consumption of farmers [36, 37]. Main authors still hesitate to see in these villae rural estates turned towards commercial monocultures, although excavations have revealed in recent decades the existence of large and small vineyards in Languedoc, Provence, and the middle Rhone Valley [10–13, 38, 39].

If the vision of an autarkic Gallo-Roman agriculture is still prominent in the historical and archaeological literature, recent syntheses nevertheless highlight the links between rural settlements and markets throughout the Roman Empire [40–42]. The Latin agronomic treatises also unambiguously describe in Italy rentier's farms clearly specialized in certain types of crops, such as olive grove and winery, of which Cato (*De Re Rustica*, X-XI) gives for example in the 2nd century BCE a particularly detailed description.

We postulate that agricultural establishments of the Gallo-Roman period practiced cash crops in addition to those intended for food crops. Nevertheless, we consider that the importance given to cash crops should be very variable from a geographical point of view. More was probably grown for markets in the areas closest to the cities than in those furthest from them, on the margins of the ancient cities.

In ROMCLIM, we did not try to model agricultural production intended to feed farmers, except in the case of cereal farms where part of the production is consumed on site. While we have no doubt that establishments oriented towards viticulture or olive growing also practiced cereal cultivation to feed farmers, we deliberately limited ourselves to modelling production intended for trade. This choice is explained by the desire to deal solely with the question of the impact of climatic variations on the Roman economy, whose main income was derived from the sale of agricultural products. Consequently, we did not try to consider productions that were not intended to be sold, even if these were of course very important for the food of the populations.

## Impact of climate variation on roman agriculture and economy: The Roman Climate Optimum (RCO) and the Late Antique Little Ice Age (LALIA)

Around the 2nd century BCE, both Latin agronomists Saserna evoked a climate change that would have made it possible to gain for the cultivation of the vine and the olive tree areas of altitude previously too cold (probably in the Apennines) to be able to practice these crops (passage quoted in Columella, *De Re Rustica*, I, 1). The reality of this significant global warming, now called the Roman Climate Optimum (RCO) or Roman Warm Period (RWP) has been highlighted by several paleoclimatic studies [2, 3]. This would have occurred as early as the middle of the 3rd century BCE and may have been particularly important. According to the results of isotopic studies carried out on sedimentary cores, the temperatures of the surface waters of the Mediterranean would have been even during the Roman period 2˚C higher than the current one [1]. The precise end of this period of climatic optimum is variably estimated between the 2nd century CE and the 4th century CE according to studies [1–3], before the beginning of the cooling of the Late Antique Little Ice Age (LALIA), paroxysmal in the 6th and 7th centuries CE [43].

Today, we wonder about the impact of these major climatic fluctuations on the societies of antiquity. In a recent book, American historian K. Harper recently argued that the RCO could have been a particularly favourable factor in the economic development of the Roman Empire during the High Empire [4]. On the other hand, it is considered that global cooling may have played a major role in the transformations of agriculture [44, 45] and the many historical and cultural upheavals of late antiquity and early Middle Ages [43].

## Agent-based modelling, archaeology and climate change

Appearing in the 1990s, simulation by agent-based modelling (ABM) is gradually gaining ground in the humanities and social sciences. ABMs are based on the programming of behavioural rules assigned to individual entities (or groups of entities) whose interactions at a micro-scale produce "emergent" phenomena at a higher level of organization (macroscopic scale). Described as a bottom-up approach, agent-based modelling is now used in various fields of fundamental and applied research, including environmental sciences, biology, human sciences (geography, sociology, archaeology), economics and other disciplines interested in the study of human and animal behaviour [46].

Presented by Tim Kohler as a "third way" for archaeology [47], agent-based modelling and the use of simulation began to develop in the 1990s in the social sciences [48]. In the United States, the disciplinary porosity between ecology and the humanities (*Human Ecology*) facilitated the transfer of this approach from one disciplinary field to another. The use of simulation for the study of the behaviour of animal societies has given archaeologists the idea of using it to form virtual human societies [49]. Models that have become iconic, such as SUGARSCAPE

[50] and ARTIFICIAL ANASAZIS [51, 52] continue to be a source of inspiration in the world of SMAs for more than twenty years.

Over the past ten years, the growing number of pedagogical manuals presenting the principles and expectations of agent-based modelling [53–56] testifies to the strongest interest in this approach in various disciplines [57]. We would like to mention a first book more specifically intended for archaeologists [58], recently published by the *Santa Fe Institute*, which promotes the use of agent-based systems in the field of social sciences and complexity from the United States.

The ability to simulate phenomena produced by interactions between entities and their environment makes ABMs particularly well suited to the study of socio-natural interactions [59]. Also, most Anglo-Saxon archaeological models are part of this paradigm to study the trajectory of ancient societies in relation to their environment, particularly on the American continent (Pueblo, Anasazi, Mayan Indians, . . .) [47, 51, 60] and in the Middle East [61]. These models are mostly devoted to a population and its territory in the prehistoric and protohistoric periods. Other rare projects are part of the long-term perspective. This is particularly the case of Medland (Mediterranean Landscape Dynamics), which focuses on the study of society-environment interactions and their impacts on landscape transformations in the Mediterranean, from the Pleistocene to the middle of the Holocene [62].

Despite the early interest of European archaeologists in agent-based models, their concrete use is not detectable in the literature until the end of the 2000s. Early models were variously developed to study salt mining in the Austrian Hallstatt mines in the Bronze Age [63], Ethnogenesis of Patagonian hunter-gatherer societies [64] or demographics, carrying capacity and the agricultural strategies of late Iron Age Celtic societies in the Czech Republic [65–68]. In France, the ANR Obresoc project (2010–2015) stimulated the design of several models to study the diffusion of Early Neolithic agriculture in Western Europe from the Near East and the Balkans [69–71].

While they were initially of interest to prehistorians and protohistorians, ABMs have also been considered in recent years with increasing interest by historians and archaeologists of antiquity in the countries of Northern Europe (Netherlands, Denmark, and England) [72, 73]. Models were thus developed in the field of Roman studies such as MERCURY, dedicated to the study of the tableware trade in the eastern part of the Roman Empire [74–76]. We can also mention ROMFARMS, designed to determine whether the Netherlands had the capacity at the beginning of our era to locally produce enough food to feed the local populations and the Roman army installed on the *Dutch* Limes [77].

Since their inception in the 1990s, ABMs have given rise to debates among modellers about the choice of principles that should govern a model. While the KISS (Keep It Simple Stupid) principle recommends programming agents as simply as possible, KIDS (Keep It Descriptive Stupid) recommend a descriptive approach that takes into account the complexity of the system from the outset [78]. In addition, practices are amended and improved thanks to the self-criticism of agent-based modelling. Opacity of functioning of the early models (« blackbox effect ») encouraged the development of standardised protocols such as Overview, Design concepts, Details (ODD), describing from the most general to the most detailed their expectations and how they work [79, 80].

The increasing complexity of the models, most of which operate stochastically, makes it increasingly difficult to understand the influence of each parameter considered on the outputs. Sensitivity analyses are now frequently used to weigh the relative importance of the various factors involved in the results of these models. A range of methods, from the simplest to the most sophisticated, using Artificial Intelligence, make it possible to better evaluate the functioning of models and their performance [81–83].

While many conceptual and methodological difficulties have been resolved over the past thirty years, others may not yet have been fully resolved. In particular, it remains difficult to validate or invalidate model outputs by the data [84], especially in archaeology, where they are often quantitatively and qualitatively insufficient. Although it may seem crucial, this final step of validating the outputs is not an absolute necessity to justify the usefulness of the models. Regardless of the significance and relevance of the results they produce, ABMs have great heuristic virtues. For archaeologists interested in phenomena and processes, they require reformulating hypotheses, collating and re-examining datasets, and forging collaborations with other disciplines [85].

It remains difficult to draw up an exhaustive inventory of agent-based models related to archaeology, as not all of them are published or even listed. Of the 1029 designs registered in the CoMSES online bookstore (comses.net)–Across all disciplines, there are currently 30 models that address archaeological issues, representing only 2.9% of the total. This observation leads us to consider that the use of agent-based modelling in archaeology is still timid compared to other disciplines.

An examination of these different models leads us to distinguish some major research themes (which are not exclusive of each other) with in particular the study of:

- Spatial phenomena: how populations, human cultures, consumer products, and epidemics have moved, expanded, and spread (e.g. [71, 86]). For these models, archaeologists favour a realistic representation of space that incorporates GIS layers. These models, known as "Spatial Agent-based models", are the most developed by archaeologists [85].

- Resource exploitation and human strategies: procurement strategies of prehistoric societies (e.g. [87–91]), agricultural strategies (Intensive/Extensive Exploitation) of protohistoric and ancient societies (e.g. [68, 77]). These approaches are strongly inspired by Site Catchment Analysis [92].

- Social phenomena: ethnogenesis, cooperation, competition, collective decision-making (e.g. [64, 93]).

- Dynamics of settlement patterns: clustering of habitats, genesis of the urban phenomenon from protohistory to historical periods (e.g. [61, 94–97]).

- Socio-environmental relations: interaction between societies, the environment and climate (e.g. [98–100]).

In general, the ABMs developed in the United States and Europe respond to various archaeological and historical issues concerning different chrono-cultural periods between Prehistory and late Antiquity, or even the beginning of the Middle Ages. For most of these models, environmental changes do not appear to be considered (perhaps because of a still insufficient involvement of paleoenvironmental specialists in these projects).

Models simulating climate variations and their impact on ancient societies are still few, although agent-based modelling appears particularly well suited for this type of problem, which is part of the complexity of human-environment relations.

## Postulates, materials and methods

### Modelling postulates and explanations

We must warn the reader that ROMCLIM simulates a roman "villa system" during a long period between the 6th c. BC and the 7th c. AD, but this exploitation system does not have historically begun in southern Gaul before the roman conquest (end of the 2nd century BC) for

collapsing at the end of Antiquity (5[th] or maybe after). So, our approach is partly ahistorical, mainly for the most part of Iron Age (6[th] c.– 2nd c. BC), but also for the 6[th] c. and 7[th] c. AD.

We do not postulate in this model that socio-economic conditions of Iron Age, Roman period (in all its duration) and Early Middle Age were the same in the area considered. They were clearly different. But we do not try to simulate the real transformations of agricultural exploitations for the different periods considered, due to the big number of uncertainties. This task remains very difficult (but this will be certainly a great challenge for a future model). By this point of view, the model is not realistic, but it's also easier to better isolate the impact of climate change, among other historical and socio-economic parameters. However, it does not mean that we consider there are no other changes driven by these other factors.

Our objective is trying to measure what could have been the profitability of the "villa system" under colder conditions (like beginning of Iron Age and early Middle Ages) than the Roman Climate Optimum (RCO) described in the text. By extending the boundaries of the Roman period to Iron Age and Late Antiquity, our main idea is to try to measure if the so-called RCO could have had (or not) an effect on roman agricultural economy. The interest of such counterfactual simulations is to assess the profits that could have been made by these ancient farms with the climate from other periods and to determine whether they would have been viable.

## Operating principles

The agents used for our ROMCLIM agent-based model (programmed in NetLogo) are family farms and wineries and olive groves. For each century in the chronological sequence explored (6[th] century BCE - 7[th] century CE), these three types of agricultural exploitations are generated by each of the cells in the model with 8 km-long sides, which corresponds to the resolution of the input climate data (for an exhaustive presentation of the model see the ODD protocol in S1 Text). Each of these exploitations produces crops annually based on potential yields that vary with the climate. These harvests are sold in the urban markets of the nearest Roman provincial capitals. Our model calculates the profits from these sales by subtracting the production and transport costs from the price paid for the crops. Wineries and olive groves remain viable as long as they are profitable; however, they disappear if their profits are zero or negative. Family farms, which operate differently, disappear when their annual harvest is insufficient to feed the whole family (6 people). Our model also calculates the total amount of potential profits for each type of exploitations according to climatic fluctuations and the values chosen by the user for their characteristics (area, number of employees), the price of the commodities, and the operating and transport costs which are variables. It is thus possible to visualize the changes in the potential profits made by the farms to ultimately discuss the impact of the climate on the agricultural economy.

## Calculation of potential yields with the LPJmL agrosystem model emulator

Potential crop yields for wheat, grapes, and olives are calculated with the ABM based on climate data and using the Lund-Potsdam-Jena managed Land (LPJmL) agroecosystem model [101, 102], which has recently been adapted to compute potential yields in the past [103–105]. Because of the large number of processes described at a daily time step and the mathematical optimization of the representation of water-carbon exchanges, the LPJmL has a high computational time cost, which usually requires the use of a computing cluster in a Linux environment (by SLURM). Therefore, in order to reproduce the functonate of LPJmL with NetLogo, we developed an emulator that simplifies this complex model through a regression equation between yields and climate (see S1 Text for the details and explanations of these equations).

For each cell of the ABM, monthly temperatures and precipitation, as well as cloud cover, are fed into the emulator, which then calculates potential grape, olive, and wheat yields. The paleoclimatic data needed to simulate yields are derived from a reconstruction performed for the Holocene at the Mediterranean scale [106] (Fig 3). These are time-slices with a 100-year time step and a spatial resolution of 5' latitude and longitude. This resolution was too low for our study, so the data were linearly interpolated at the 8 km scale. We have decided not to work at a finer resolution due to calculations limitations with NetLogo.

## Calculation of agricultural production and its market value

The olive and wine farms were simulated according to the characteristics given by the Latin agronomist Cato (2nd century BCE). In his agronomic treatise, he describes a vineyard of 100 iugera (25 ha) which the exploitation required 16 people, or 1.5 person/ha (*De Agricultura*, XI). He also accurately describes a 240-iugera (60 ha) olive grove for which 13 people were needed (*De Agricultura*, X). The ratio here is 0.21 people/ha, which is seven times less workforce than for a vineyard.

For each century considered, the olive and wine farms produce an average annual harvest of wine and oil based on the area farmed and the potential yields returned by the LPJmL emulator. While for wine, the production is directly calculated in hectoliters by the model, a conversion is applied to the weight of the olives to convert it into liters of oil. The number of kilos of olives needed to produce a liter of oil depends on the varieties grown (5–8 kg), but in our study we used the lowest value of 5 kg/l, the figure most often proposed in the archaeological literature for the Roman period [107].

The market value of the production is then calculated by the model according to a market price selected by the user in a range between 0 and a maximum value, which varies according to the commodity. The Edict of Diocletian (301 CE), which established a maximum price for all agricultural products in the Roman Empire, set prices from 2 to 30 denarii/sextarius for wine (1 sextarius = 1/2 liter) and from 8 to 40 denarii/sextarius for oil, depending on the quality.

Grain farms are operated by a family of 6 people, according to the theoretical model of a family farm of the Roman period [33], which can also be applied to the Iron Age. The area cultivated by these farms is set by the user up to a maximum of 6 ha, which corresponds to the maximum area that a family is capable of farming. The Saserna (2nd century BCE father and son agronomists) argued that a single man could cultivate 8 iugera (2 ha) by hand. If we consider a family composed of 2 adults, 2 elderly people, and 2 children, we can estimate that the "workforce" cannot be based on more than 3 people (i.e., 6 ha).

The farms produce an average annual wheat crop based on the area farmed and the potential yield. Part of this harvest is destined to feed the family, based on an annual ration of 200 kg/person. Another portion is saved to sow the fields the following year at a rate of 5 modius/ iugerum (135kg/ha) according to Columella's recommendations (*De Re Rustica*, II, 9). The remaining surplus of grain is sold at a variable market price. Historians generally retain values of 2, 2.5, or 3 sestertii/modius [41, 108]. 100 denarii/modius was the maximum price set by Diocletian's Edict (1 modius = 8.67 liters).

## Production and transport costs, calculating profits

The production costs of vineyards and olive farms are proportional to the number of people employed, at a rate of 140 sestertii/person/year, according to a proposed estimate for the annual cost of a slave or free employee [109]. The cost of transporting goods, also taken into account, is calculated according to the distance separating each farm from the nearest

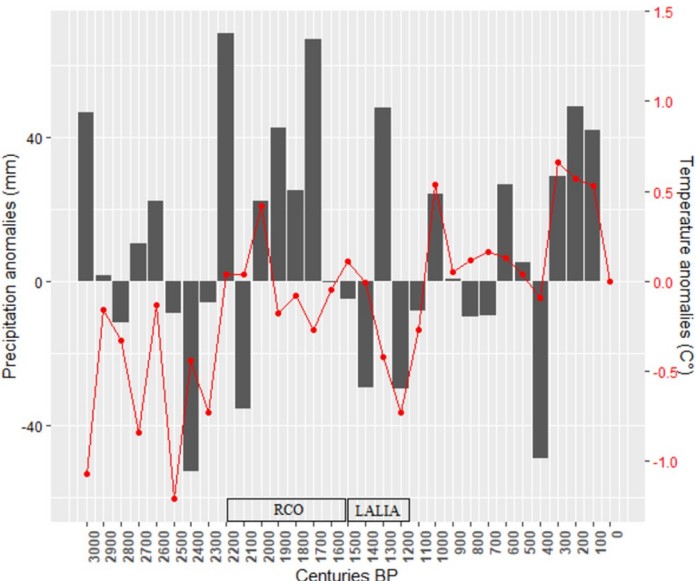

**Fig 3. Trend in annual average temperature (red curve) and precipitation (gray bars) anomalies for each century of the last three millennia compared to the present in Southern France (data from Guiot and Kaniewski 2015, point 55).** RCO means Roman Climate Optimum LALIA means Late Antique Little Ice Age.

provincial capital and the price of transporting the goods by road. In our model, an agricultural exploitation sells its products at the nearest urban markets, which was not in reality always the case. For the maximum price of transporting goods by land, we take the rate set by the Diocletian's Edit of 20 denarii/mile/1200 pounds (i.e., 0.14 sestertii/km/kg).

Finally, the annual profit for each type of farm is calculated by the model by subtracting production and transportation costs from the market price of the output.

## Results

To determine the sensitivity of the outputs of our model to the different parameters considered, we performed a Sensitivity Analysis with the One-Factor-at-Time (OFAT) method (see S2 Text). We have thus varied each parameter one by one while keeping the others at constant value. The results of this analysis show that the different parameters do not have the same importance for each type of crop. If climatic factors (precipitation, temperatures) are preponderant in the model for viticulture, they appear a little less influential for olive growing. Grain cultivation, on the other hand, appears to be much more sensitive to economic parameters (market prices, transport costs) than to climate.

Fig 4 presents the results obtained with the model using the OFAT method used for the Sensitivity Analysis. The graphs show for each parameter and each type of crop the potential benefits calculated by the model. The reference curves (in black) show the results obtained with the median values of the four parameters. Colored envelopes include all the results that can be obtained between the minimum and maximum of the range of values of each of them. Changing the settings of the parameters makes it possible to obtain different potential benefits for each crop and to accentuate more or less the trends of evolution, which nevertheless remain similar.

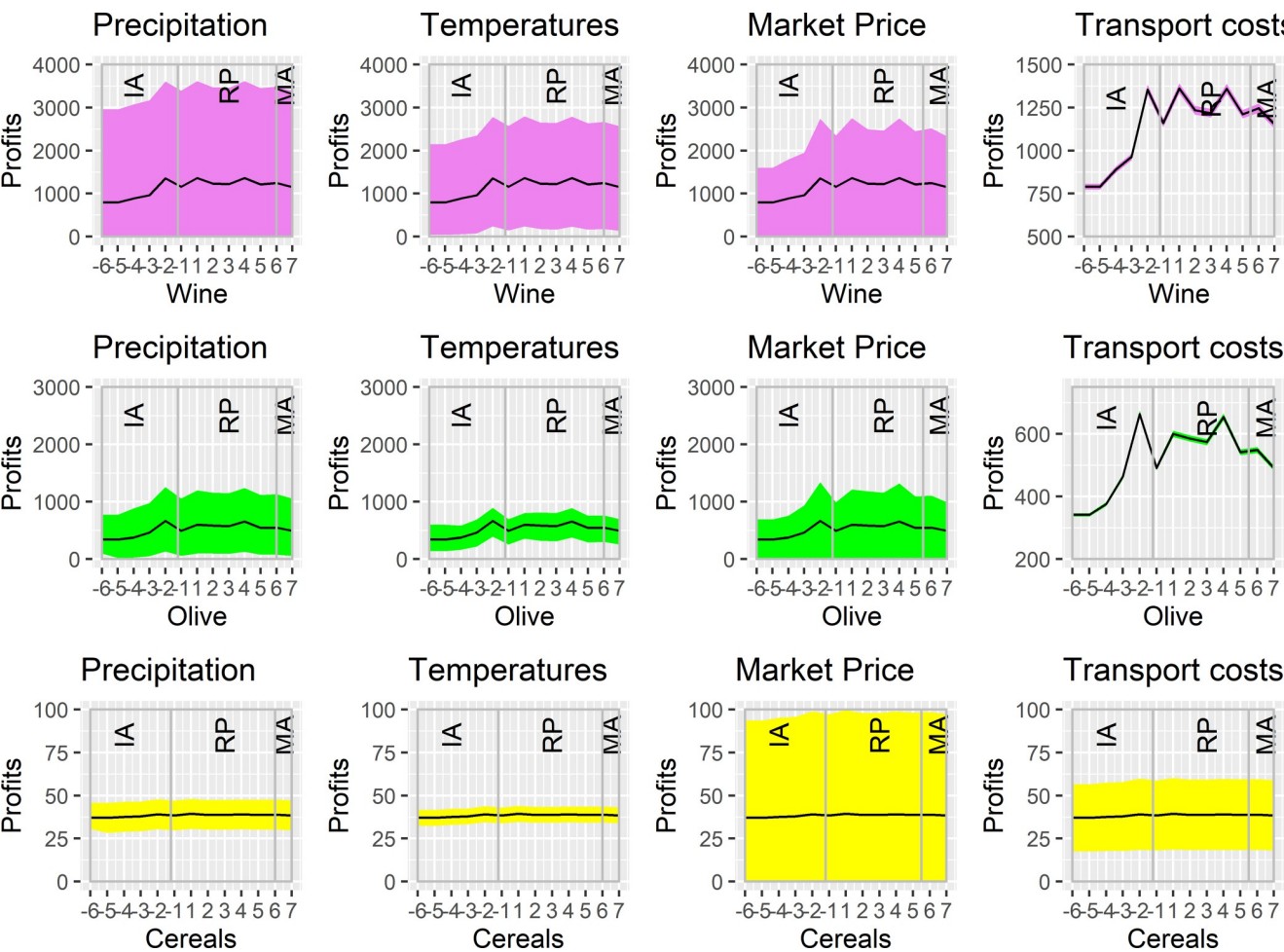

**Fig 4. Graphs showing the potential benefits obtained for viticulture, olive growing and cereal cultivation according to the variations in the values of each parameter considered in the model (precipitation, temperatures, market prices and transport costs).** The x-axis represents the chronological sequence, and the y-axis represents the potential benefits (in millions of sestertii).

### Temporal trend in potential benefits

Fig 4 also shows the trend in the cumulative potential profits by type of farm (olive, wine, grain) for each century in the sequence. We observe that the potential profits generated by the wineries are clearly higher than those of the olive groves, but also of the grain farms, which generate the lowest profits. The profit curves of wineries and olive groves are parallel to each other. Their secular fluctuations are very significant, especially for wine, unlike those of the grain farms, which have a much flatter curve. These results are indirect evidence of the greater sensitivity of vine and olive yields to climatic variations (in the LPJmL model emulator), as compared to wheat.

The curves start at their lowest level in the 6[th] century BCE, the coldest century according to the paleoclimatic reconstruction used. From this point onwards, a constant secular increase is observed for vineyards and olive farms during the Iron Age until the 2[nd] century BCE, which marks a first peak before a significant inflection of the curves in the 1[st] century BCE. Potential profits then reach a high level again during the High Empire (1[st]-3[rd] century CE) and then decrease during late antiquity (4[th]-7[th] century CE).

## Changes in the geography of profitable zones

The potential profit maps produced by the model for the time sequence studied show more or less significant differences according to the centuries considered. For the 6[th] century BCE (Fig 5), the potential profits generated by viticulture are the most significant in the Languedoc plains, in the middle Rhone valley, and in the Antibes and Fréjus region. Olive growing profits change much more subtly, with higher values to the north-east of Nîmes, in the Var, and in Roussillon, about 20 km southwest of Perpignan. Some regions have no potential for either olive or wine production, such as the hinterland of Languedoc and the High Alps, where only wheat cultivation yields modest profits (Fig 5).

If one compares these results with those produced for the 1[st] century CE, for example (Fig 6), the profits are much greater for viticulture but also for olive growing, which appears to be extremely profitable in the Languedoc hinterland. This result deserves to be emphasized insofar as this potential was non-existent with the climatic characteristics of the 6[th] century BCE.

## Discussion

### Strengths and uncertainties of the model

**What real densities of agricultural exploitations? What spatial coverage?.**   In ROM-CLIM, each cell representing an area of 64 km² is exploited by three agricultural exploitations (a vineyard, an olive farm, a cereal farm). If this density (0.04 exploitations/km²) may seem too low to be realistic, it is nevertheless much higher than those we can calculate from archaeological data. To take the example of the Var department - where research has been particularly intensive - there are 103 olive/wine-growing farms for an area of 5973 km² [110]. The density of 0.01 archaeological sites/km² is here therefore significantly lower than that simulated in the model. Of course, archaeological data represent only a greater or lesser fraction of reality. The densities calculated with these data should be considered minimums. It is still difficult to put forward credible estimates for the density of agricultural settlements in Gallia Narbonensis, but the densities simulated in ROMCLIM - four times higher than those we can apprehend by archaeology - cannot be considered low.

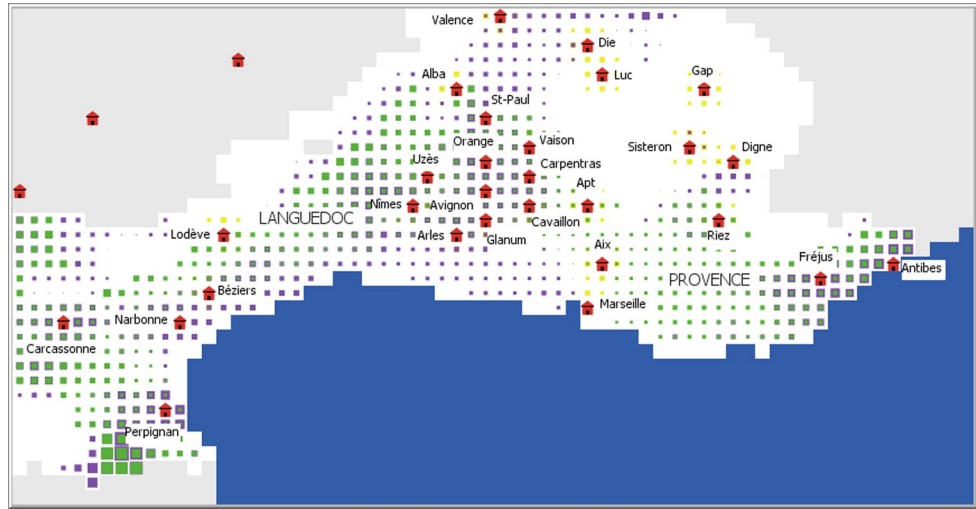

**Fig 5. Map of potential profitability of villae and farms with climate characteristics of the 6[th] century BCE.** The size of the symbols is proportional to the profitability of the wine (purple squares), olive (green squares), and grain (yellow squares) farms. The red symbols represent the provincial capitals during the Roman period.

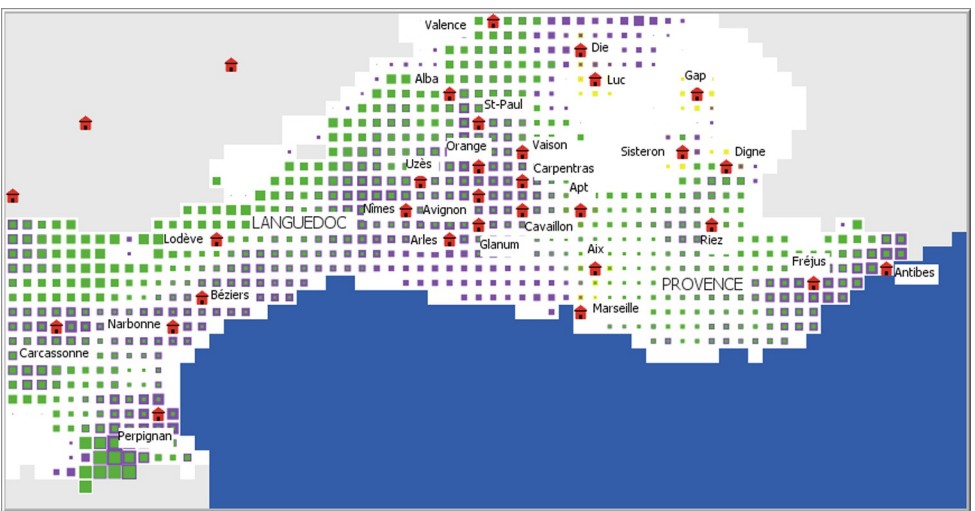

**Fig 6. Map of potential profitability of villae and farm with climate characteristics of the 1st century CE.** The size of the symbols is proportional to the profitability of the wine (purple squares), olive (green squares), and grain (yellow squares) farms. The red symbols represent the provincial capitals during the Roman period.

On the other hand, we carried out the simulations with virtual agricultural exploitations evenly distributed throughout the study area. However, archaeological data suggest that these exploitations were concentrated in certain areas (on the outskirts of agglomerations) to leave others vacant. The potential benefits calculated at the output of the model by simulating full exploitation of the study area may therefore be overestimated, considering that commercial agriculture was maybe concentrated in only a few privileged areas.

**Potential predictive value of the outputs.** But by performing "full-gauge" simulations, the model has the advantage of producing results in areas where archaeological data are scarce or absent, which gives the model a potentially predictive power for the location of agricultural exploitations and the geography of crops. If the landowners of antiquity acted with a form of economic rationality (which remains a subject debated among historians) it can indeed be assumed that they sought to optimize their profits by creating or buying their wineries, olive groves and cereal farms in the most favourable places for each of these crops from a geographical, environmental, and climatic point of view, in other words areas defined by our model.

For viticulture, it is observed that the model designates as potentially favourable a geographical area much larger than that identified by archaeology in the middle Rhone Valley. For olive growing, the high potential displayed in the Languedoc hinterland is also not corroborated by archaeological data, but research has been not very intensive in this sector. The results could therefore have a predictive value that could encourage new research (we will develop this point farther).

## Outlooks for improving the calculation of yields for cereals?

Regarding questions about the vulnerability and resilience of ancient societies to climate change, one of the major interests of the model is to be able to simulate the impact of changes in temperature and precipitation on yields and agricultural productivity using the LPJmL emulator. Simulations show that while the climatic factor is important, it does not have the same impact depending on the type of crop.

The high sensitivity of viticulture and olive growing to variations in rainfall and temperature highlighted by the model supports the hypothesis of a strong impact of climate change on

ancient economy, still largely based on agricultural income. The less pronounced effect of these changes on cereal cultivation, on the other hand, suggests a potentially moderate effect on societies and their demography, considering that cereals were the basis of the diet.

In another hand, the importance of grain in Gallia Narbonensis is still certainly underestimated by archaeological and historical sources. If ROMCLIM does not simulate the potential production of cereal by the villae due to the lack of historical information, the model shows that wheat produced by farms would be however the only profitable crop in the Alps. In mountain areas that are too cold for wine and olives, wheat would have been the only possible source of income among the different cash-crops.

However, our model may underestimate these climate effects on grain farming. The cereal yields calculated are based on a parameterization of LPJmL which groups and averages the biological and phenological characteristics of barley and naked wheat (the two main cereals grown in Southern Gaul). It should be noted, however, that different species and varieties of cereals do not respond to climate change in the same way. But in the future, we plan to develop new specific parameterizations for barley, naked wheat, and other cultivated cereals in LPJmL to test the reaction of each of them to changes in precipitation and temperature (ongoing ANR MICA project). The results obtained could help to qualify our conclusions on the relatively moderate impact of climate change on grain farming.

## Historical questions and models outcomes

In general, the results returned by our model show that the potential profits of the virtual agricultural exploitations gradually rise during the Iron Age, under the effect of a warmer and wetter climate, and reach a maximum during the High Empire, before declining during late antiquity (Fig 4). Overall, these results are consistent with the hypothesis of a beneficial effect of the Roman Climate Optimum on the economy of the Roman Empire [4], but also that of the negative impact of climate on agricultural yields in late antiquity [43, 111]. Our modelling therefore clearly highlights the role of the climate in fluctuations in yields and agricultural exploitations profitability, particularly for vine and olive cultivation, while grain farming appears to be less sensitive to these variations.

**The positive effects of the Roman Climate Optimum (RCO).** Our ROMCLIM model provides a good account of the significant effects of climate change on the potential yields of the crops and the profitability of the agricultural exploitations. The upward slope of the curves in the 3rd century BCE (Fig 4) reflects climatic conditions that were more favourable than in previous centuries for grape and olive yields. However, it is during this same century that the archaeological and archaeobotanical data mentioned above attest to the development of these crops in Southern Gaul [7, 16]. This development could therefore have been favored by warmer but also wetter climatic conditions, which is all the more likely since some studies place the beginning of the RCO as early as 250 BCE [3].

However, one could be surprised that villae were not more widespread in Gallia Narbonensis as of the 1st century BCE, because this territory had been integrated into the Roman empire and ipso facto into its market economy since the end of the 2nd century BCE. According to the results produced by ROMCLIM, for the 1st century BCE, there was a clear downturn in the potential profitability of vineyards and olive groves, which reflects less favourable climatic conditions than in the previous century (Fig 4). Nevertheless, the results obtained remain superior to those of the 3rd century BCE, which does not allow us to assert that this secular climatic variation was of sufficient magnitude to delay the development of villae.

The answer to this question could therefore be historical and economic. In the 1st century BCE, Cicero (*De Republica*, III, 6) asserted that the Romans sought to limit the cultivation of

olives and grapevines beyond the Alps to increase the value of their wines and olives. However, the actual existence of Roman protectionist laws remains a controversial topic among historians [17]. The reasons why the cultivation of commercial crops in Southern Gaul in the 1st century BCE was still undeveloped are not yet elucidated.

**The negative impact of the Late Antique Little Ice Age (LALIA).** The decrease in yields and potential profits from grapevine and olive cultivation starting in the 4th century CE indicates a loss of potential profitability of the farms. These results support the hypothesis of a negative impact of LALIA on commercial agriculture and the economy during late antiquity. On the other hand, the potential yields and profits of grain cultivation appear to be only slightly affected. The effects of climatic cooling on a wheat-based diet are therefore less obvious here. While an increase in crop failure under LALIA (AD 450–750) is now being discussed [4, 43, 44], our model results suggest that the effects of late antique climatic cooling may not have had a particularly dramatic effect on agriculture in Southern Gaul. But we have discussed upper of a potential issue with the sensitivity of cereals yields to the climate in the LPJmL model. So, we need further investigations to comfort these conclusions.

## Was olive growing highly developed during antiquity in the Languedoc hinterland?

If we compare the profitability maps produced by the ROMCLIM model with the archaeological data concerning the grapevine and the olive tree, we notice a certain geographical concordance of the results for the High Empire. The strong potential highlighted by our model in the Languedoc plains, the middle Rhone valley, and Provence for viticulture is supported by the results of archaeological research which have highlighted the importance of this crop [9, 11, 14, 112]. While olive growing at this time has been identified in the Var [13, 112], our model does not show a particularly strong potential in this department. Meanwhile, the potential profitability of olive farms in the Languedoc hinterland was high (Fig 6).

In Languedoc, the profitability map produced by ROMCLIM for the High Empire shows a remarkable similarity with a land-use model proposed thirty years ago [19]. Based on archaeological data, this model contrasted a coastal strip where viticulture was practiced with a hinterland (below the present-day limit of the olive tree) where olives were cultivated. Although the discoveries made by preventive archaeology have largely confirmed the pre-eminence of viticulture in the Languedoc plains, the reality of olive growing at this time in the hilly hinterland has been questioned. The identification of parts of a press that were discovered there as being intended for oil production has been disputed [12]. It is therefore believed today that this activity was only limited compared to Provence [112].

The high potential profitability of olive farms in Gallia Narbonensis, simulated by ROMCLIM, may therefore seem surprising in relation to these arguments. However, if we lend credence to the description by Sidonius Apollinaris (*Epistula* II, 9, 1) of the hills in the Nîmes hinterland planted with grapevines and olive trees, olive growing was more or less developed in this area in the 5th century BCE [13]. However, there is almost no undisputable trace of this activity in today's archaeological data. But, in the hinterland areas, field research (surveys, excavations) is generally more difficult and much less intensive than in the coastal plains, which also benefit from the development of preventive archaeology. The results provided by ROMCLIM could therefore lead to a serious reassessment of the importance of olive growing during the Roman period in the Languedoc hinterland.

## Conclusion

The results produced by our ROMCLIM agent-based model support the hypothesis of a very positive effect of the Roman Climate Optimum in Southern Gaul on the yields and profitability of agricultural exploitations, which then decline during the Late Antique Little Ice Age (LALIA). For the entire chronological period studied between the middle of the first Iron Age and the early Middle Ages, the highest potential farm incomes calculated by the model are reached during the High Empire (Fig 4). This finding is consistent with the archaeological investigations in Southern Gaul, which have shown that the commercial cultivation of grape-vines, and to a lesser extent the olive tree, reached their peak during this period [112]. Global warming could therefore had seriously boosted the rise of the estate economy within the framework of the villa system developed in our early era. In a pre-industrial society in which most of the wealth still came from the land, a significant increase in the profitability of estates linked to an increase in yields due to a favourable change in climate (to which grapevines and olive trees are particularly sensitive), was certainly more globally a powerful engine for economic growth not only in Gaul but also in the Roman Empire [4].

Yet the effects of the Roman Climate Optimum were probably felt in Southern Gaul well before the beginning of the Roman period, in the middle of the second Iron Age. Our ROM-CLIM simulation clearly shows the beginning of a simulated trend of increasing profits from vineyards and olive farms as early as the 3rd century BCE (Fig 4). While this result is anachronistic and counterfactual (the Roman farms we simulated in our program did not yet exist), it does indirectly indicate a significant increase in potential yields. As already mentioned previously, archaeological and paleoenvironmental surveys conducted in Southern Gaul suggest an increase in olive and wine growing in the second half or towards the end of the 3rd century BCE [7, 16]. While the nature of the socio-economic and historical factors behind this phenomenon has been questioned, an explanation based on the climate should now be considered more seriously given the results of our modelling. Palynological analyses have already concluded that the Roman Warm Period was established as early as 250 BCE in northwest Spain [3]. Roman commercial agriculture could therefore certainly have developed in Gaul rapidly after the Roman conquest in the 120s BCE, at a time when the climate was already favourable for profitable villa farms. The fact that this speculative agriculture apparently did not take off before the end of the 1st century BCE is therefore probably linked to historical or socio-economic factors that are still difficult to identify.

The geographical concordance of the results produced by ROMCLIM and the field data for the different crops studied can be interpreted as proof that our model is quite accurate, at least for the High Empire. However, it is still difficult to fully judge the relevance of our results for the entire geographical area concerned because of the still diffuse nature of the archaeological data from a spatial point of view. In the sectors for which the model indicates a good potential profitability of the farms, but where these concrete data are still rare or absent, we consider that the ROMCLIM results have a potential predictive and heuristic value. The simulated high profitability of olive farms in the Languedoc hinterland during the Roman period could, for example, lead to a reassessment of the controversial question of the importance of olive farming in this area and to additional research.

In general, our model highlights that global warming (with moisture) allowed for strong economic growth as of the moment that commercial agriculture combined with the villa system developed before in Italy [113] was established in Southern Gaul at the beginning of our era. The climatic and socio-economic conditions were then particularly favourable to the profitability and development of viticulture, which was the driving force behind the economic development of Southern Gaul, ahead of olives and wheat. Without this climatic change, it is

not certain that the Roman Empire's economy would have been able to develop so significantly.

From a methodological point of view, this research shows the potential of such an approach - an agent-based model combined with an agroecosystem model - for other crops and/or other periods and/or other ancient societies.

Even if this approach cannot take in account all the complexity of the socio-ecological processes, it at least allows us to test out various hypotheses. It enables us to shed light on these hypotheses that emerge from the data through mechanistic processes. Such work requires an intrinsically interdisciplinary approach.

On the methodological and heuristic basis of this first work, we could improve this model in the future by taking in account more historical and socio-economic factors and replace for example the regional area considered in a wider system (i.e. Roman Empire), and the commercial interactions with other provinces.

## Supporting information

**S1 Text. ODD (Overview Design Details) protocol.**
(DOCX)

**S2 Text. Sensitivity analysis.**
(DOCX)

**S3 Text. References for SI.**
(DOCX)

**S1 Table. State variables of agricultural units.**
(DOCX)

**S2 Table. State variables of grid cells.**
(DOCX)

**S3 Table. Seasonal climate variables.**
(DOCX)

**S4 Table. Prices given by the Edict of Diocletian (301 AD).**
(DOCX)

**S5 Table. Minimum, average, and maximum values of each parameter used for the sensitivity analysis (SA).**
(DOCX)

**S1 Fig. Example of January precipitation and temperature fields interpolated for 2500 and 2000 yr BP (VIth and Ist century CE) in the South of France.**
(TIF)

**S2 Fig. XY-plot of yields provides by LPJmL and the emulator.**
(TIF)

**S3 Fig. Tornado diagrams presenting the results of the sensitivity analysis for wine, olive oil and cereal production.** The parameters are ranked from top to bottom from most to least important. The data on the abscissa represent the sum of the benefits (in millions of sestertii) generated by all virtual agricultural exploitations.
(TIF)

**S1 File. The NetLogo model ROMCLIM with the other files are available in GitHub https://github.com/Bernigaud2021/ROMCLIM.**
(TXT)

## Acknowledgments

We are grateful to Charles La Via for editing the English, and to the two anonymous reviewers. Their remarks and suggestions had led to improve the structuration of the paper and the output analysis of the model.

We acknowledge too the valuable inputs of the RDMed team during numerous upstream discussions: Alan Kirman (CAMS, EHESS, Paris), Sander Van der Leeuw (Arizona State University's School of Sustainability, United States), Sylvain Olivier (University of Nîmes, France), Daniel Contreras (University of Florida, United States), Gül Sürmelihindi (University of Mainz, Germany) and Cees W. Passchier (University of Mainz, Germany).

## Author Contributions

**Conceptualization:** Nicolas Bernigaud.

**Investigation:** Nicolas Bernigaud, Joël Guiot, Philippe Leveau, Delphine Isoardi.

**Methodology:** Nicolas Bernigaud, Alberte Bondeau, Joël Guiot, Frédérique Bertoncello, Marie-Jeanne Ouriachi, Loup Bernard.

**Project administration:** Joël Guiot.

**Supervision:** Joël Guiot.

**Validation:** Nicolas Bernigaud, Alberte Bondeau, Joël Guiot, Frédérique Bertoncello, Marie-Jeanne Ouriachi.

**Writing – original draft:** Nicolas Bernigaud.

**Writing – review & editing:** Alberte Bondeau, Joël Guiot, Frédérique Bertoncello, Marie-Jeanne Ouriachi, Laurent Bouby, Loup Bernard, Delphine Isoardi.

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
