## [Decision Letter · Decision Letter 0]

4 May 2023

PONE-D-23-05108The impact of climate change on the agriculture and the economy of Southern Gaul, New perspectives of agent-based modelingPLOS ONE

Dear Dr. Bernigaud,

Thank you for submitting your manuscript to PLOS ONE. After careful consideration, we feel that it has merit but does not fully meet PLOS ONE’s publication criteria as it currently stands. Therefore, we invite you to submit a revised version of the manuscript that addresses the points raised during the review process. Neither of the original reviewers was available  to review your revised manuscript. Each of the new reviewers provides comments and suggestions to improve your manuscript. Please address all of the comments and suggestions while making your revisions.

We look forward to receiving your revised manuscript.

Kind regards,

John P. Hart, Ph.D.

Academic Editor

PLOS ONE

Journal Requirements:

4. We note that Figures 1 and 2 in your submission contain map images which may be copyrighted. All PLOS content is published under the Creative Commons Attribution License (CC BY 4.0), which means that the manuscript, images, and Supporting Information files will be freely available online, and any third party is permitted to access, download, copy, distribute, and use these materials in any way, even commercially, with proper attribution. For these reasons, we cannot publish previously copyrighted maps or satellite images created using proprietary data, such as Google software (Google Maps, Street View, and Earth). For more information, see our copyright guidelines: http://journals.plos.org/plosone/s/licenses-and-copyright.

 a.          You may seek permission from the original copyright holder of Figures 1 and 2 to publish the content specifically under the CC BY 4.0 license. 

Reviewers' comments:

Reviewer's Responses to Questions

**Comments to the Author**

1. Is the manuscript technically sound, and do the data support the conclusions?

Reviewer #1: Yes

Reviewer #2: Partly

2. Has the statistical analysis been performed appropriately and rigorously? 

Reviewer #1: Yes

Reviewer #2: No

3. Have the authors made all data underlying the findings in their manuscript fully available?

Reviewer #1: No

Reviewer #2: Yes

4. Is the manuscript presented in an intelligible fashion and written in standard English?

Reviewer #1: Yes

Reviewer #2: Yes

5. Review Comments to the Author

Reviewer #1: had a fresh read of the paper without looking at other reviewers' comments; base approach: if it is good then it is good. no need to artificially add points to the authors' workload.

1. Downloaded and executed your model. Why does one need to choose a directory if there is a standard location ("External Files")? Also, C:/Users/Bernigaud Nicolas/Dropbox/RDMed_postdoc/Article_RDMed_2021/Soumission_PNAS/ROMCLIM_data and so forth literally in the model => replace by .

2. grapes_rainfed_hlha_ag_2500BP_LScor.asc not found => include in External Files ; this might not be the only file missing, so please check carefully

3. "The impact of climate change on the agriculture and the economy of Southern Gaul, New perspectives of agent-based modeling" suggest using a double point instead of a comma

4. abstract: well written

5. introduction: well written

6. the study area: well written and necessary for getting a background on your subject, nothing to be desired more.

7. state-of-the-art: does not include documentation (https://www.jasss.org/23/2/7.html) even though this comes later; would have expected a (by-)sentenence in the ABM part

8. Calculation of potential yields: "in order not to lose the interactivity made possible by NetLogo," ... however from a simulation perspective this interactivity during a run leads to non-reproducable results, please discuss this in your discussion section; I understand it from an explorative viewpoint, though. "we developed an LPJmL emulator that simplifies the operation of this complex model through a regression equation between yields and climate"  details missing, a shame! Others could benefit from your approach, if you'd be so kind and describe it please? Also this is another contribution of your submission which you could make explicit (e.g. put a sentence that you also contribute that into the introduction)

9. Write NETLOGO or NetLogo (latter one is preferred) but not both.

10. calculation of agricultural production - workforce required - have I missed the demographics model that must be somehow running in the background?

11. results - nothing to add or take away, perfect as it is

12. discussion - additional factors such as "international" trade could be added (future perspectives); else perfectly discussed

13. conclusion: is a bit big, but anyway. good as it stands.

Reviewer #2: Review on “The impact of climate change on the agriculture and economy of Southern Gaul, new perspectives of agent-based modelling”

General Comments

The paper presents an interesting question by relating agricultural yields with environmental changes and market benefit. It does so by employing Agent-Based Modelling in a context where these have already proven to be very useful for archaeological inference. I consider valuable the emulation of the LPJmL emulator (although it might need further detailing) and the capacity to infer potential yields based in environmental constrains.

There would be, however, some questions to be considered. The paper assumes a similar trade/market (and thus socio-economic and demographic) structure for very different chronological periods such as the 6th century BCE, before the Roman conquest and the 7th century CE, after the Roman fall. Is it safe to assume the same structure for such different periods (including all the lapse of the Roman Empire in the region)?

Additionally, the paper is in perhaps in general too deterministic. Several of the values offered are uncertain (e. g. Lines 274-295, but also environmental factors). In this case, the authors generally opt to model the means, but perhaps addressing these issues from a probabilistic perspective (perhaps in further works) could offer a better nuance for the problem. One example of the use of uncertainty, also applying ABMs to the Roman period would be, for example, Carrignon et al. 2022.

Specific Comments

- Line 53: I am not sure about these references. Since this is the introductory section, and the first time where ABMs have been presented, perhaps, maybe references to seminal papers, both general and specific to Roman Archaeology (e. g. Axtell et al. 2002; Brantingham 2003 or then Lake, 2014) would be more appropriate. The references mentioned in the paper could be included as specific examples in the more extended explanation of ABMs, within the state-of-the-art section.

- Line 57: What does “geography of cultures” means exactly in this particular context? Could you please develop?

- Lines 175-177: In this sense, perhaps it would be necessary to provide an assessment of how much agriculture was destined for trade and how much was destined for self-feeding. This is because the proportions destined for one another will have an effect on the land patches considered. It would not the same considering them all for trading use, as the ABM seems to do, than considering some percentage of those land patches as ‘not available’ for trading. Alternatively, it would reasonable to think that, if there is correlation between environment and agricultural trading, this should also account for the fact that years with poor harvest should result in an exponential reduction of trading resources because, in this case, trading would not be affected just by the poor harvests themselves (less production), but also by the detraction of agricultural goods for self-feeding, since self-feeding agriculture would have also had poorer payoffs. Depending on the magnitude of autarchy, these are potential bias factors that should be accounted for in the model. This could be done without too much difficulty by introducing the concept of a hierarchical random effect where the level could be, for example, different regions or different climatic niches.

- Line 189: Please, change the word ‘bunch’ by another more appropriate to the academic register.

- Line 194: Reference(s) would be needed after “according to studies”.

- Line 202: In this section, the authors offer an overview of ABMs. However, they do not explain what they are, and perhaps a (not necessary long) explanation would be useful for the reader not used to these types of models.

- Line 211: References 60 and 61 are incorrect. They are cited as the full book, and although these books do contain chapters with ABMs, both of them have several chapters where simulation is performed by other means and, in the case of 61, some of these chapters are not even based on simulation strictly speaking. Additional, when discussing ABMs applied to prehistory there would be several important references missing here (e. g. works by Crema, Lake, Barton and several others)

- Line 213. Please, substitute ‘England’ by the ‘United Kingdom’.

- Lines 233-234. Related to my comment above. Differently to wineries and olive groves, in the model family farms disappear when they cannot feed six people because “harvest is insufficient”. This indeed implies that the family is actually being fed from their own farm and, consequently, and considering six people as an average family unit, only the surpluses could be put to the market or, in other words, the equivalent of food for six people should be detracted from total amount of food sent to the market. Is this being done by the model? Additionally, if this assumes that farms will feed the families running them, why should this not be the case for olive groves and wineries?

- Lines 257-258. When faced with the limitations of NetLogo, the authors could consider continuing their analysis with other languages, such as Python or R.

- Lines 352-354. How can the authors discard that these profits are due to the better climatic conditions, and not to general socio-economic conditions of the Empire at this time?

On the ODD

- It would be nice to have some additional review on the English in this section.

- The explanation of the regression for the potential yields is a bit messy. I understand what is happening, but I believe it would be good if the authors could clearly state, in standard mathematical language, what they are doing? Also, I understand they are showing the coefficients of the parameters under a linear modelling perspective, but the linear model producing those coefficients has not been shown, nor its residuals, applicability, etc. This would be necessary to understand whether these coefficients work, since they are the basis for the paper.

6. PLOS authors have the option to publish the peer review history of their article (what does this mean?). If published, this will include your full peer review and any attached files.

Reviewer #1: No

Reviewer #2: No

---

## [Author Response · Author response to Decision Letter 0]

8 Nov 2023

Journal Requirements:

4. We note that Figures 1 and 2 in your submission contain map images which may be copyrighted. All PLOS content is published under the Creative Commons Attribution License (CC BY 4.0), which means that the manuscript, images, and Supporting Information files will be freely available online, and any third party is permitted to access, download, copy, distribute, and use these materials in any way, even commercially, with proper attribution. For these reasons, we cannot publish previously copyrighted maps or satellite images created using proprietary data, such as Google software (Google Maps, Street View, and Earth). For more information, see our copyright guidelines: http://journals.plos.org/plosone/s/licenses-and-copyright.

Authors-> Figures 1 and 2 were made by the main author (N. Bernigaud) with R and the package rnaturalearth. (https://www.naturalearthdata.com/about/terms-of-use/). So, all the data used (vector and raster) are in public domain. None copyrighted maps or other documents were used.

Response to Reviewers:

Reviewer #1: had a fresh read of the paper without looking at other reviewers' comments; base approach: if it is good then it is good. no need to artificially add points to the authors' workload.

1. Downloaded and executed your model. Why does one need to choose a directory if there is a standard location ("External Files")? Also, C:/Users/Bernigaud Nicolas/Dropbox/RDMed_postdoc/Article_RDMed_2021/Soumission_PNAS/ROMCLIM_data and so forth literally in the model => replace by .

Authors-> Right, we hadn’t chosen the useful option indeed. So, we had replaced in the code section the former path by the location of the folder “External files” in GitHub (https://github.com/Bernigaud2021/ROMCLIM/tree/main/External_files)

Changes in the code in GitHub can be seen at: https://github.com/Bernigaud2021/ROMCLIM/blame/main/ROMCLIM2021.nlogo

2. grapes_rainfed_hlha_ag_2500BP_LScor.asc not found => include in External Files ; this might not be the only file missing, so please check carefully

Authors->Thanks, in this case, the import of this file wasn’t necessary for this published version of ROMCLIM (it was for an older), so we have disabled it in the code. We have rechecked all the files. All is OK now.

3. "The impact of climate change on the agriculture and the economy of Southern Gaul, New perspectives of agent-based modeling" suggest using a double point instead of a comma

Authors->Better indeed. Done.

Reviewer #1: 4. abstract: well written

Authors->Thanks.

Reviewer #1: 5. introduction: well written

Authors->Thanks.

Reviewer #1: 6. the study area: well written and necessary for getting a background on your subject, nothing to be desired more.

Authors->Thanks.

Reviewer #1: 7. state-of-the-art: does not include documentation (https://www.jasss.org/23/2/7.html) even though this comes later; would have expected a (by-) sentence in the ABM part.

Authors->OK, better indeed. We have added in the section “state of the art” sentences to introduce the ODD protocol (see lines 258-262).

8. Calculation of potential yields: "in order not to lose the interactivity made possible by NetLogo," ... however from a simulation perspective this interactivity during a run leads to non-reproducable results, please discuss this in your discussion section; I understand it from an explorative viewpoint, though.

Authors->OK, in fact the first interest for emulating LPJmL was not the possibility to change (randomly or not) the values of the parameters during an emulation. So, the sentence we wrote was maybe a little bit inaccurate. So, we change it. We have created an emulator which simplified the functioning of LPJmL, because it wasn’t possible to reproduce and program it with NetLogo in all its complexity. One advantage of this simplification is a greater rapidity of calculation, which facilitates the process of simulation (see lines 354-357).

"we developed an LPJmL emulator that simplifies the operation of this complex model through a regression equation between yields and climate" details missing, a shame! Others could benefit from your approach, if you'd be so kind and describe it please?

Authors-> The LPJmL emulator works with three equations (for vine, olive production & cereals), which could be found with their coefficients in the ODD (see Annex S1, 7). The location of this information wasn’t indeed mentioned in the main text. This is now the case (see lines 358-359).

Also this is another contribution of your submission which you could make explicit (e.g. put a sentence that you also contribute that into the introduction)

Authors->OK, useful indeed. We have added a sentence in the introduction (see lines 59-61)

9. Write NETLOGO or NetLogo (latter one is preferred) but not both.

Authors-> True, lack of harmonization. So, we have opted for “NetLogo” everywhere.

10. calculation of agricultural production - workforce required - have I missed the demographics model that must be somehow running in the background?

Authors-> This model doesn’t integrate a demographic model (it could be). We have considered “Workforce” for grain farm as only 3 persons in a family of 6, and 2 ha a maximum of land that could be cultivated by one person. So, the maximum area cultivated is fixed to 6 ha per farm (lines 397-399).

11. results - nothing to add or take away, perfect as it is.

Authors-> Thanks.

12. discussion - additional factors such as "international" trade could be added (future perspectives); else perfectly discussed.

Authors-> Yes, we perfectly agreed that commercial agricultural activity at regional scale is strongly dependent of its place/location in a wider system: here the Roman Empire. As well noticed by the reviewer, this parameter (as others) is not taken in account in this (first) version of the model. So, we have added some sentences about this point at the end of the conclusion part for future development (see lines 677-670).

13. conclusion: is a bit big, but anyway. good as it stands.

Authors-> Thanks!

Reviewer #2: Review on “The impact of climate change on the agriculture and economy of Southern Gaul, new perspectives of agent-based modelling”

General Comments

Reviewer #2: The paper presents an interesting question by relating agricultural yields with environmental changes and market benefit. It does so by employing Agent-Based Modelling in a context where these have already proven to be very useful for archaeological inference. I consider valuable the emulation of the LPJmL emulator (although it might need further detailing) and the capacity to infer potential yields based in environmental constrains.

There would be, however, some questions to be considered. The paper assumes a similar trade/market (and thus socio-economic and demographic) structure for very different chronological periods such as the 6th century BCE, before the Roman conquest and the 7th century CE, after the Roman fall. Is it safe to assume the same structure for such different periods (including all the lapse of the Roman Empire in the region)?

Authors-> To avoid some misunderstandings with the readers, we must better explain our approach. We don’t postulate in this model that socio-economic conditions of Iron Age, Roman period (in all its duration) and Early Middle Age were the same in the area considered (southern Gaul). They were different. We don’t try to simulate the transformations of agricultural exploitations between Iron Age and Early Middle Age. Considering the big number of uncertainties, this task remains very difficult (but this will be certainly a great challenge for another model).

So, the objective of ROMCLIM is to simulate (under different climate condition) a roman “villa system”, enough documented by historical texts and archaeology. This so-called “villa system” don’t have historically begun in southern Gaul before the roman conquest (end of the 2nd century BC) for collapsing at the end of Antiquity (5th or maybe after). So, our approach is partly anachronistic and/or counterfactual, mainly for the most part of Iron Age (6th c. – 2nd c. BC), but also for the 6th c. and 7th c. AD. This is certainly an unusual (maybe a little bit disturbing) perspective, mainly for most of archeologists and people paying attention to the facts, but we must make understand it: our objective is trying to measure what could have been the profitability of the “villa system” under colder conditions (like beginning of Iron Age and early Middle Ages) than the Roman Climate Optimum (RCO) described in the text. By extending the boundaries of the Roman period to Iron Age and Late Antiquity, our main idea is to try to measure if the so-called RCO could have had (or not) an effect on roman (agricultural) economy. The model doesn’t change the structure of agrarian system between the different period (this is not realistic, and this point could be criticized), mainly because the task seems too difficult. But it’s also easier to better isolate the impact of climate change, among other historical and socio-economic parameters. However, it doesn’t mean that we consider there are no other changes driven by these other factors.

For a better understanding and clarification of our approach, we have added a paragraph (“Modelling postulates and explanations) in the main text (see lines 312-332).

Reviewer #2: Additionally, the paper is in perhaps in general too deterministic.

Authors-> We perfectly understand that our approach can be perceived (unfortunately) as “deterministic”, but we want to nuance this judgment. In fact, we have tried to avoid this pitfall in environmental sciences (by the point of view of social sciences specialists) by integrating -beside climate data- some socio-economic factors (social organization of rural exploitations, costs of production & transport,…). In the conclusion part, we write that climate change influenced Roman economy, but in conjunction with historical factors (see lines 656-657). However, the influence of climate change and socio-economic factors were tested in the Sensitivity Analysis (see S2). This analysis shows that climate factors have an influence on the outputs of the model, but not with the same intensity for the different crops. The weight of climate factor was not predetermined in entry but tested. So, if this paper sounds “deterministic”, we hope that is not truly the case for the reasons explained here.

Reviewer #2: Several of the values offered are uncertain (e. g. Lines 274-295, but also environmental factors). In this case, the authors generally opt to model the means, but perhaps addressing these issues from a probabilistic perspective (perhaps in further works) could offer a better nuance for the problem. One example of the use of uncertainty, also applying ABMs to the Roman period would be, for example, Carrignon et al. 2022.

Authors-> Several values are uncertain and/or variable in space and time. For some parameters, we have indeed kept a mean or single value (ratio of seeds, quantity of food consumed by person), considering the variations of these parameters should affect slightly the results of the model. But for most of the parameters considered we have taken in account a range of values (and not even one). All the possible results which can be obtained with these different values are presented in the Sensitivity Analysis for temperatures, precipitations, market prices and transport costs (see S2). So, we really have explored a range of possibilities in regard of parameters variations and uncertainties. But we agree that things could be improved or differently tested (for a further model) by using probabilistic method/Bayesian statistics as developed for example in Carrignon et al. 2022 (now mentioned in line 253).

Specific Comments

Reviewer #2: - Line 53: I am not sure about these references. Since this is the introductory section, and the first time where ABMs have been presented, perhaps, maybe references to seminal papers, both general and specific to Roman Archaeology (e. g. Axtell et al. 2002; Brantingham 2003 or then Lake, 2014) would be more appropriate. The references mentioned in the paper could be included as specific examples in the more extended explanation of ABMs, within the state-of-the-art section.

Authors-> Agree. We have considered that splitting the references for ABMs in two parts (in introduction, then in the part I.4) was finally not a satisfying solution. So, we have removed the references in the introduction and put all of them in the I.4 part (lines 208-310), which was totally rewritten and extent. All the references suggested by the reviewer were integrated into.

Reviewer #2: Line 57: What does “geography of cultures” means exactly in this particular context? Could you please develop?

Authors-> By “geography of cultures”, we mean the patchwork of regions individualized by the nature of the dominant crop (some are mainly specialized in wine production, other in olive oil, and other in cereals for example…). But indeed, the expression “geography of cultures” seems to be inaccurate or confused in English. So, we have replaced it by “geography of crops” (see lines 58 and 599), which is certainly more correct and intelligible.

Reviewer #2: Lines 175-177: In this sense, perhaps it would be necessary to provide an assessment of how much agriculture was destined for trade and how much was destined for self-feeding. This is because the proportions destined for one another will have an effect on the land patches considered. It would not the same considering them all for trading use, as the ABM seems to do, than considering some percentage of those land patches as ‘not available’ for trading.

Authors-> This ABM simulates only the commercial crops indeed. We have explained and justify in the main text why we don’t take in account self-feeding production (too difficult task, lack of data, but also our questions about the link between climate and monetarized economy,…) (see lines 178-185). We can state that’s it’s very difficult (nor impossible) to assess how much surface of land was dedicated to commercial agriculture and how many was dedicated to self-feeding agriculture. This relative proportion was variable in space and time and difficult to estimate, even roughly. In the beginning of our era, southern Gaul was largely dedicated to commercial agriculture, if we give credence to Strabo, who describes the landscape of Narbonensis dominated by wine and olive culture (i. e. commercial productions) (see lines 105-107). But this historical description is obviously too vague for weighting the importance of commercial agriculture, certainly not practiced with the same intensity in the Gallia Narbonensis. So, the only solution will be to use a random function and observe the results with numerous simulations (but see our response lower).

Alternatively, it would reasonable to think that, if there is correlation between environment and agricultural trading, this should also account for the fact that years with poor harvest should result in an exponential reduction of trading resources because, in this case, trading would not be affected just by the poor harvests themselves (less production), but also by the detraction of agricultural goods for self-feeding, since self-feeding agriculture would have also had poorer payoffs.

Authors-> Climatic “bad years” impact indeed negatively agricultural production in general, commercial agriculture and self-feeding production. Of course, in the case of a succession of very “bad years” for cereals, it could be supposed that it was difficult to feed more globally the population (starving episode) and the employee/slaves of commercial exploitations like wineries & olive groves. We must admit that these cumulative effects are not totally taken in account in ROMCLIM (but this improvement will be possible in an update version of the model). However, the model takes partly in account this aspect, insofar as farmer who are producing grain can be impacted in their own subsistence (and disappear in our model), if the production of cereals is too low to feed themselves for climatic reasons (but this is an extreme scenario). But the resilience of the agricultural system in Roman period was certainly important, considering that a (relative rich) owner of wineries and olive grove can buy on the market (in the whole Roman empire) some stock of cereals produce elsewhere to feed his employees. So, a complete collapse of the agricultural production system should be considered as an exceptional event.

Depending on the magnitude of autarchy, these are potential bias factors that should be accounted for in the model. This could be done without too much difficulty by introducing the concept of a hierarchical random effect where the level could be, for example, different regions or different climatic niches.

Authors-> Before the publication, we have sought for a solution in which we consider than patches/location where villae are known by archaeology were mainly dedicated to agricultural production (> 50 %), and those where there is no data were cultivated for self-feeding. But this solution which restrain the commercial agriculture in some area, according to archaeological data, is not satisfying for us. We believe indeed that these archaeological data underestimated the boundaries of commercial agricultural landscape.

So, in a modelling approach, we could use a random function to give a different ratio in each patch. It’s arbitrary and difficult to justify in an historical point of view, but worthy in an explorative perspective. The idea advanced by the reviewer (the use of a hierarchical random effect) is truly stimulating. We hadn’t sought about that before. So (in a future model now), we could consider the ratio as a variable, lead a big number of simulations and use Bayesian statistics: it’s a promising perspective.

Reviewer #2: Line 189: Please, change the word ‘bunch’ by another more appropriate to the academic register.

Authors ->Well, inappropriate indeed. We have replaced “bunch” by “several” (line 194).

Reviewer #2: Line 194: Reference(s) would be needed after “according to studies”.

Authors -> OK, done (line 194).

Reviewer #2: Line 202: In this section, the authors offer an overview of ABMs. However, they do not explain what they are, and perhaps a (not necessary long) explanation would be useful for the reader not used to these types of models.

Authors ->It will be more useful indeed with a little bit more explanations. We have rewritten and extended (a lot) this part (see lines 208-310).

Reviewer #2: Line 211: References 60 and 61 are incorrect. They are cited as the full book, and although these books do contain chapters with ABMs, both of them have several chapters where simulation is performed by other means and, in the case of 61, some of these chapters are not even based on simulation strictly speaking. Additional, when discussing ABMs applied to prehistory there would be several important references missing here (e. g. works by Crema, Lake, Barton and several others)

Authors ->Right, books mentioned are not fully dedicated to ABMs. Former references 60 and 61 were removed. As suggested, we have replaced these references by others more appropriate in history of ABMs.

Reviewer #2: Line 213. Please, substitute ‘England’ by the ‘United Kingdom’.

Authors -> OK, fine. Done.

Reviewer #2: Lines 233-234. Related to my comment above. Differently to wineries and olive groves, in the model family farms disappear when they cannot feed six people because “harvest is insufficient”. This indeed implies that the family is actually being fed from their own farm and, consequently, and considering six people as an average family unit, only the surpluses could be put to the market or, in other words, the equivalent of food for six people should be detracted from total amount of food sent to the market. Is this being done by the model?

Authors -> Yes, the model truly calculates the surplus by subtracting from the annual production of grain the quantity necessary to feed the family (on the ratio of 200kg/person/year) but also the quantity of seed needed for the next year (these details were explained at the end of the ODD in S2).

Reviewer #2: Additionally, if this assumes that farms will feed the families running them, why should this not be the case for olive groves and wineries?

-> Authors: The cases of wineries and olive groves are different from cereal farms, as they are exclusively rentiers exploitations. The only goal of these exploitation is to gain money. However, it’s indeed probable that a part of wine or olive oil they produced were consumed by the personal. But historical sources don’t give any quantitative indication about that (only the payment of slave with cereals for feeding). For wineries and olive groves, we postulate (but this can be discussed) that the consummation of oil and wine by the staff was negligible, relative to the global amount of production.

Reviewer #2: - Lines 257-258. When faced with the limitations of NetLogo, the authors could consider continuing their analysis with other languages, such as Python or R.

-> Authors: the limitations invocated by us are maybe less imposed by NetLogo itself, than the hardware we used and its computing power (desktop computer). However, the use of Python and R are to consider indeed.

Reviewer #2: Lines 352-354. How can the authors discard that these profits are due to the better climatic conditions, and not to general socio-economic conditions of the Empire at this time?

-> Authors: In our conclusion we wrote that the rise of economy in roman period probably result from the spread of a commercial agriculture (the “villa system”) in conjunction with better yields, boosted by an increase of temperature and precipitation during the Roman Climate Optimum (see lines 653-658). So, we don’t conclude of our results that better climatic conditions are the exclusive or main factor of the rise (and fall) of the roman agriculture, but a significant one among possible others. As already mentioned, our model doesn’t simulate the Roman socio-natural system and changes in socio-economic conditions in all its aspects and complexity (this can be criticized, of course). In this version, we are focusing mainly on the impact of climate change on yields, agricultural production, and profits. It should be necessary of course to take in account more parameters to better weighting them. This opens the path for future improvements of the model.

On the ODD

Reviewer #2: It would be nice to have some additional review on the English in this section.

-> Authors: OK, English in the ODD was reread, corrected, and improved.

Reviewer #2: The explanation of the regression for the potential yields is a bit messy. I understand what is happening, but I believe it would be good if the authors could clearly state, in standard mathematical language, what they are doing? Also, I understand they are showing the coefficients of the parameters under a linear modelling perspective, but the linear model producing those coefficients has not been shown, nor its residuals, applicability, etc. This would be necessary to understand whether these coefficients work, since they are the basis for the paper.

-> Authors: The emulator consists in 3 linear equations (for vine, olive and cereals) with multiple coefficients calculated by the regression of outputs of LPJmL on seasonal climate data use in entry. For each crop, the result is a linear equation with 8 coefficients. For a more detailed presentation of the LPJmL emulator we have improved the ODD (See S1, 7). We add the variance for each equation and a figure which shows the comparison of the emulator and the simulations done by LPJmL on all the time slices.

---

## [Decision Letter · Decision Letter 1]

23 Jan 2024

PONE-D-23-05108R1The impact of climate change on the agriculture and the economy of Southern Gaul: New perspectives of agent-based modelingPLOS ONE

Dear Dr. Bernigaud,

Thank you for submitting your manuscript to PLOS ONE. After careful consideration, we feel that it has merit but does not fully meet PLOS ONE’s publication criteria as it currently stands. Therefore, we invite you to submit a revised version of the manuscript that addresses the points raised during the review process. The reviewer is positive about your revisions in response to their original review. They have identified some very minor corrections that need to be made to the text. Once you have addressed these issues, I will be happy to accept the paper for publication. *PLOS ONE* does not provide page proofs. So, you may want to do a final, thorough proofread before making your final submission.

We look forward to receiving your revised manuscript.

Kind regards,

John P. Hart, Ph.D.

Academic Editor

PLOS ONE

Journal Requirements:

Reviewers' comments:

Reviewer's Responses to Questions

**Comments to the Author**

1. If the authors have adequately addressed your comments raised in a previous round of review and you feel that this manuscript is now acceptable for publication, you may indicate that here to bypass the “Comments to the Author” section, enter your conflict of interest statement in the “Confidential to Editor” section, and submit your "Accept" recommendation.

Reviewer #2: All comments have been addressed

2. Is the manuscript technically sound, and do the data support the conclusions?

Reviewer #2: Yes

3. Has the statistical analysis been performed appropriately and rigorously? 

Reviewer #2: Yes

4. Have the authors made all data underlying the findings in their manuscript fully available?

Reviewer #2: Yes

5. Is the manuscript presented in an intelligible fashion and written in standard English?

Reviewer #2: Yes

6. Review Comments to the Author

Reviewer #2: The authors have addressed most of my concerns, both in the responses and in the text. As I understand now, these were more due to how I understood the text, rather than to problems in the text itself. In any case, the clarifications included at the relevant parts of the text might help other readers to understand better the extent and objectives of the model.

On a particularly positive note, the historiography on ABMs applied to archaeology has greatly improved both in terms of content and references. At this stage, I would only have some small notes to add, for the authors to consider.

Line 28. Please, change ‘process’ by ‘processes’

Line 186. Please, change ‘agronomists’ by ‘agronomist’

Line 187. Please, consider changing “gain to” by “gain for”.

Line 305. Please, consider changing “suited to” by “suited for”.

Line 310. Please, consider changing “doesn’t” by “does not”, more appropriate to the academic register. In general, please substitute all the abbreviations by their non-abbreviated counterpart when applicable.

Line 354. Parallelisation and multicore computation can be done in any OS. Although I myself am a Linux user and supporter, why would Linux be a requirement? Perhaps it is not necessary to mention this at all in the text, but if the authors do want to reference it, perhaps refer more broadly to customary use on UNIX/Slurm on computer clusters?

7. PLOS authors have the option to publish the peer review history of their article (what does this mean?). If published, this will include your full peer review and any attached files.

Reviewer #2: No

---

## [Author Response · Author response to Decision Letter 1]

31 Jan 2024

Reviewer #2: The authors have addressed most of my concerns, both in the responses and in the text. As I understand now, these were more due to how I understood the text, rather than to problems in the text itself. In any case, the clarifications included at the relevant parts of the text might help other readers to understand better the extent and objectives of the model.

On a particularly positive note, the historiography on ABMs applied to archaeology has greatly improved both in terms of content and references. At this stage, I would only have some small notes to add, for the authors to consider.

Line 28. Please, change ‘process’ by ‘processes’.

Authors-> Thanks, done.

Line 186. Please, change ‘agronomists’ by ‘agronomist’.

Authors-> The Saserna agronomists were both: so, the plural is accurate.

Line 187. Please, consider changing “gain to” by “gain for”.

Authors-> Thanks, done.

Line 305. Please, consider changing “suited to” by “suited for”.

Authors-> Thanks, done.

Line 310. Please, consider changing “doesn’t” by “does not”, more appropriate to the academic register. In general, please substitute all the abbreviations by their non-abbreviated counterpart when applicable.

Authors-> Thanks, done.

Line 354. Parallelisation and multicore computation can be done in any OS. Although I myself am a Linux user and supporter, why would Linux be a requirement? Perhaps it is not necessary to mention this at all in the text, but if the authors do want to reference it, perhaps refer more broadly to customary use on UNIX/Slurm on computer clusters?

Authors-> The developers of the LPJmL model work in a Linux environment because they estimate it’s the more practical choice (even if now other solutions in other OS are good). But we mention more explicitly SLURM in line 354.

---

## [Editor Report · Decision Letter 2]

1 Feb 2024

The impact of climate change on the agriculture and the economy of Southern Gaul: New perspectives of agent-based modeling

PONE-D-23-05108R2

Dear Dr. Bernigaud,

We’re pleased to inform you that your manuscript has been judged scientifically suitable for publication and will be formally accepted for publication once it meets all outstanding technical requirements.

Kind regards,

John P. Hart, Ph.D.

Academic Editor

PLOS ONE
---

## [Editor Report · Acceptance letter]

29 Feb 2024

PONE-D-23-05108R2 

PLOS ONE

Dear Dr. Bernigaud, 

I'm pleased to inform you that your manuscript has been deemed suitable for publication in PLOS ONE. Congratulations! Your manuscript is now being handed over to our production team.

Kind regards, 

on behalf of

Dr. John P. Hart 

Academic Editor

PLOS ONE